# Concurrent Reactive Oxygen Species Generation and Aneuploidy Induction Contribute to Thymoquinone Anticancer Activity

**DOI:** 10.3390/molecules26175136

**Published:** 2021-08-25

**Authors:** Mohammed Al-Hayali, Aimie Garces, Michael Stocks, Hilary Collins, Tracey D. Bradshaw

**Affiliations:** 1Al-Mosul General Hospital, Mosul University Post Office, P.O. Box 11104, Mosul 41002, Iraq; 2School of Pharmacy, Biodiscovery Institute, University Park, University of Nottingham, Nottingham NG7 2RD, UK; aimie.garces@nottingham.ac.uk (A.G.); michael.stocks@nottingham.ac.uk (M.S.); hilary.collins@nottingham.ac.uk (H.C.)

**Keywords:** thymoquinone, aneuploidy, apoptosis, ROS generation, GSH depletion

## Abstract

Thymoquinone (TQ) is the main biologically active constituent of *Nigella sativa*. Many studies have confirmed its anticancer actions. Herein, we investigated the different anticancer activities of, and considered resistance mechanisms to, TQ. MTT and clonogenic data showed TQ’s ability to suppress breast MDA-MB-468 and T-47D proliferation at lower concentrations compared to other cancer and non-transformed cell lines tested (GI_50_ values ≤ 1.5 µM). Flow-cytometric analyses revealed that TQ consistently induced MDA-MB-468 and T-47D cell-cycle perturbation, specifically inducing pre-G1 populations. In comparison, less sensitive breast MCF-7 and colon HCT-116 cells exhibited only transient increases in pre-G1 events. Annexin V/PI staining confirmed apoptosis induction in MDA-MB-468 and HCT-116 cells, which was continuous in the former and transient in the latter. Experiments revealed the role of reactive oxygen species (ROS) generation and aneuploidy induction in MDA-MB-468 cells within the first 24 h of treatment. The ROS-scavenger NAD(P)H dehydrogenase (quinone 1) (NQO1; DT-diaphorase) and glutathione (GSH) were implicated in resistance to TQ. Indeed, western blot analyses showed that NQO1 is expressed in all cell lines in this study, except those most sensitive to TQ-MDA-MB-468 and T-47D. Moreover, TQ treatment increased NQO1 expression in HCT-116 in a concentration-dependent fashion. Measurement of GSH activity in MDA-MB-468 and HCT-116 cells found that GSH is similarly active in both cell lines. Furthermore, GSH depletion rendered these cells more sensitive to TQ’s antiproliferative actions. Therefore, to bypass putative inactivation of the TQ semiquinone metabolite, the benzylamine analogue was designed and synthesised following modification of TQ’s carbon-3 atom. However, the structural modification negatively impacted potency against MDA-MB-468 cells. In conclusion, we disclose the following: (i) The anticancer activity of TQ may be a consequence of ROS generation and aneuploidy; (ii) Early GSH depletion could substantially enhance TQ’s anticancer activity; (iii) Benzylamine substitution at TQ’s carbon-3 failed to enhance anticancer activity.

## 1. Introduction

Thymoquinone (TQ, Figure 1) is the major bioactive phytochemical of the plant *Nigella sativa*. Many researchers describe TQ’s ability to inhibit cancer cells’ growth via targeting signalling pathways that underpin cancer hallmarks [1,2,3], e.g., proliferative signalling, growth suppressor evasion, resisting cell death, and metastasis [4]. For instance, TQ suppresses PI3K/Akt signal transduction in MDA-MB-468 and T-47D breast [5], induces p53/p21 expression in HCT-116 colon, and induces caspase-dependent apoptosis in A549 lung carcinoma cells [6]; it also inhibits bone metastasis of breast MDA-MB-231 cells [7]. Moreover, TQ can be used as an adjuvant agent that sensitises cancer cells to chemotherapy by suppressing resistance mechanisms [8,9]. Effenberger et al. (2010) found that TQ improved doxorubicin potency against leukaemic HL-60 cells and enhanced sensitivity of multi-drug resistant MCF cells to topotecan [10].

Reactive oxygen species (ROS) production by natural product or natural product-derived chemotherapy agents (such as doxorubicin) is known to contribute to anticancer activity [11]. Free radicals are known mediators of DNA damage through oxidising nucleoside bases, leading to cell-cycle arrest, and either repair or apoptosis [12]. TQ acts as antioxidant (at low concentration) and pro-oxidant (at higher concentration) [13]. Interestingly, TQ can also serve as a pro-oxidant even at low concentrations in the presence of copper ions, which co-exist at considerably higher levels in various malignancies [13,14]. Indeed, TQ’s cellular oxidative damage leading to apoptosis is described by other researchers, e.g., in lymphoma [15], colon [16], hepatic [17], ovarian [18], and prostate [19] carcinoma cells. However, TQ’s oxidative stress can be neutralised by cellular-protective ROS scavengers: NAD(P)H: quinone oxidoreductase 1 (NQO1) and glutathione (GSH) [20]. For instance, NQO1 confers protection for MCF-7 against TQ’s oxidative stress, and its inhibition by dicoumarol renders MCF-7 cells TQ-sensitive [21]. Furthermore, GSH depletion enhanced TQ-induced apoptosis in human laryngeal HEp-2 carcinoma cells [22].

Herein, we investigated the activity and resistance mechanisms of carcinoma cells to TQ. First, we confirmed TQ’s ability to inhibit proliferation and induce apoptosis in carcinoma cells with different sensitivities. We then examined TQ’s ability to induce specific early cell-cycle perturbations in sensitive cells and possible ROS contribution to these changes. Thirdly, we interrogated the correlation between ROS scavengers and cancer cells’ sensitivity to TQ, and whether GSH depletion could modulate these sensitivities. Guided by ROS-generation and -scavenger results, we synthesised a structural analogue of TQ in an attempt to bypass resistance mechanisms and determine whether such modification can enhance its antiproliferative activity compared to TQ. 

## 2. Results

### 2.1. Thymoquinone Inhibits Growth and Colony Formation of Carcinoma Cells

TQ significantly suppressed the growth and colony-formation of cancer cell lines. As shown in MTT assays (Figure 2A, Table 1), TQ potently inhibited MDA-MB-468 and T-47D proliferation, with low GI_50_ values of ≤1.5 µM. In comparison, TQ’s GI_50_ values in A549 and HT-29 were higher, reaching ~19 and ~27 µM, respectively. HCT-116, MCF-7, and MIAPaCa-2 cells were similarly sensitive to TQ, with GI_50_ ~ 12 µM. MRC-5 (lung fibroblasts) were used as a normal cell line model, revealing a GI_50_ value of 13.4 μM. DMSO vehicle did not affect cancer cell growth (Appendix A). In clonogenic assays, TQ significantly suppressed carcinoma cell colony formation (Figure 2B and Appendix A). The average plating efficiencies were: A549, 30%; HCT-116, 22%; HT-29, 30%; MCF-7, 25%; MDA-MB-468, 20%; MIAPaCa-2, 30%; and T-47D, 35%. MDA-MB-468 and T-47D cells exhibited the highest sensitivities to TQ, with significant inhibition of colony formation at ~1 µM, while no colonies were seen at 5 µM. Other cell lines were less sensitive, and colony formation was suppressed at concentrations ≥ 5 µM. 

### 2.2. Thymoquinone Caused Perturbation of Cell-Cycle Progression

To determine whether the antiproliferative action of TQ was a consequence of cell-cycle perturbations, flow-cytometric cell-cycle analysis was implemented, using PI to stain the DNA. Four cell lines were selected: the more sensitive being MDA-MB-468 and T-47D and the less sensitive being MCF-7 and HCT-116. The concentrations used were guided by MTT and clonogenic assay results: for MDA-MB-468, T-47D: 1, 5, 10 μM; and for MCF-7, HCT-116: 10, 20 μM. The most significant cell-cycle perturbation seen in MDA-MB-468 and T-47D cells after TQ exposure was a concentration-dependent increase in pre-G1 events, indicating apoptosis induction (Figure 3 and Appendix A). MDA-MB-468 showed decreased events in G1, especially after 72 h exposure to 1 and 5 μM TQ. T-47D exhibited decreased G1 and G2/M events at 5 μM after 72 h. At 10 μM, cells from both cell lines appeared to have undergone massive apoptosis accompanied by a stark decrease in G1, S, and G2/M events after 24 and 72 h exposures. MCF-7 cells also exhibited significant concentration-dependent increases in pre-G1 events accompanied by decreased G1 and S phases, especially following 24 h exposure to 20 μM TQ. After 72 h, MCF-7 populations appeared to have recovered from TQ-induced perturbation at 10 μM. However, 72 h exposure to 20 μM TQ was still able to induce significantly increased pre-G1 and decreased G1 and S events, but to a reduced extent compared to 24 h. Likewise, in HCT-116 cells, 24 h exposure to 20 μM TQ revealed significant accumulation of pre-G1 events with decreased G1, S, and G2/M events, while no change was seen with 10 μM TQ. Following 72 h, 10 μM TQ induced significant G1/S arrest with decreased G2/M events; at 20 μM, the most significant perturbation (though less than after 24 h treatment) was the accumulation of pre-G1 events (Figure 3 and Appendix A).

### 2.3. Thymoquinone Induced Apoptosis

To confirm that pre-G1 events detected in cell-cycle analyses were due to apoptosis induction, annexin V/PI flow-cytometry-based apoptosis assays were carried out in two selected cell lines: MDA-MB-468 and HCT-116. TQ concentrations adopted were 1, 5, and 10 μM for MDA-MB-468 and 10 and 20 μM for HCT-116 cells. For MDA-MB-468, consistent with the pre-G1 population, annexin V/PI showed a concentration-dependent increase in annexin-positive (A+) cells (Figure 4 and Appendix A). Following 24 h treatment, the majority of apoptotic events detected were in the early-apoptotic quadrant, giving ~15% for 1 µM, 25% for 5 µM, and 45% for 10 µM TQ. After 72 h, apoptotic events had shifted to the late-apoptotic quadrant, yielding 11% for 1 µM, 19% for 5 µM, and 72% for 10 µM TQ. In HCT-116 cells, TQ induced significant concentration-dependent increased apoptosis following 24 h exposure, with the majority of apoptotic events being in the late-apoptotic quadrant (A+/PI+), reaching 10% for 10 µM and 23% for 20 µM. However, following 72 h, reduced and non-significant increases in apoptotic cells were seen in these cells even with 20 µM treatment (Figure 4 and Appendix A). Similarly, fewer pre-G1 events were indicated in HCT-116 cell-cycle analyses after 72 h compared to 24 h treatment.

### 2.4. Thymoquinone Induced Reactive Oxygen Species Generation

Quinones are known ROS inducers, so we hypothesised that TQ can generate ROS, contributing to apoptosis induction. To test this hypothesis, ROS levels were measured following 6 and 24 h exposures to TQ (1, 5, 10 μM) in both MDA-MB-468 and T-47D cells (Figure 5). After 6 h, TQ significantly increased ROS in both cell lines concentration-dependently. ROS levels reached ~1.2-fold for 1 μM, ~2.0-fold for 5 μM, and ≥4-fold for 10 μM TQ compared to untreated control. Following 24 h, reduced ROS generation was seen, approaching 1.2- and 1.5-fold of untreated control for 5 and 10 μM TQ, respectively. However, no increases in ROS levels were seen in 1 μM-treated MDA-MB-468 and T-47D cells following 24 h. 

### 2.5. Thymoquinone Induced Early Temporary Aneuploidy in MDA-MB-468 Cell Cycle

The early, significant ROS generation suggested that cancer cell proliferation (cell cycling) may be perturbed as early as within 6 h exposure of MDA-MB-468 cells to TQ. We therefore inspected MDA-MB-468 cell-cycle distribution after 6, 12, and 24 h exposure to 5 μM TQ (Figure 6). After 6 h, cell-cycle histograms indicated that TQ induced significant early-onset perturbations including the appearance of aneuploid cells at G1, S, and G2/M phases. Following 12 h, a stark increase in aneuploid cells was seen in all cell-cycle phases, accompanied by increased pre-G1 events. After 24 h, the percentage of aneuploid cells decreased to lower levels compared to 6 and 12 h treatment groups, while some pre-G1 events could still be detected. No aneuploid populations were observed in the untreated samples.

### 2.6. Thymoquinone Induced NQO1 in Carcinoma Cells

Cancer cells can exploit ROS scavengers to outmanoeuvre increased ROS levels. Hence, we hypothesised that TQ-induced ROS triggered expression and/or activity of ROS scavengers, e.g., NQO1 and GSH. To confirm this hypothesis, NQO1 expression was investigated by western blot in untreated A549, HCT-116, HT-29, MCF-7, MDA-MB-468, MIAPaCa-2, and T-47D protein lysates (Figure 7). MDA-MB-468 and T-47D did not show any detectable NQO1 expression; however, obvious NQO1 bands were detected in lysates prepared from all other cell lines. Band intensities, determined in arbitrary units by densitometry/image studio software, are represented graphically in Figure 7. Next, we checked whether TQ was able to increase NQO1 expression in two selected carcinoma cell lines (MDA-MB-468 and HCT-116) (Figure 8). HCT-116 cells were treated with 10 and 20 μM TQ and MDA-MB-468 cells with 5 and 10 μM TQ for 24 and 72 h. In HCT-116, TQ treatment significantly increased NQO1 expression in a concentration-dependent manner. After 24 h, TQ (10 and 20 μM) increased NQO1 expression in HCT-116 cells by 3- and 10-fold respectively; following 72 h, 3- and 7-fold enhanced expression was detected using 10 and 20 μM respectively. Conversely, NQO1 expression was not detected in lysates of MDA-MB-468 cells irrespective of prior exposure to TQ. 

### 2.7. Thymoquinone Depleted GSH in HCT-116 and MDA-MB-468

Several quinones are known to deplete cellular GSH [23]; hence, we tested whether TQ was able to deplete GSH. GSH activity was measured in HCT-116 and MDA-MB-468 following 24 h TQ treatment using 1 and 5 μM (MDA-MB-468) and 10 and 20 μM (HCT-116) TQ (Figure 9A,B). TQ (5 μM) significantly depleted GSH activity in MDA-MB-468 cells to ~40% of the control values. TQ (20 μM) decreased GSH activity in HCT-116 to ~50% of untreated control. However, no significant change in GSH activity was seen using lower concentrations. 

### 2.8. GSH Depletion Enhanced Thymoquinone Antiproliferative Activity

GSH (in addition to NQO1) has been implicated in resistance to TQ. Therefore, we hypothesised that early GSH depletion of carcinoma cells before TQ introduction might prolong oxidative stress, rendering these cells more sensitive to TQ antiproliferative effects. To test this hypothesis, we selected buthionine sulfoximine (BSO) as a GSH-depleting agent. BSO reduces levels of glutathione by inhibiting γ-glutamylcysteine synthetase, the enzyme required in the first step of glutathione synthesis and being investigated as an adjunct in cancer treatment, to increase cancer cells’ sensitivity to oxidative stress caused by chemotherapy [24]. BSO by itself could also affect cell viability. Accordingly, we aimed to achieve maximum GSH depletion with minimal effect on cell viability. Initially, MTT assays were conducted to determine the sensitivity of MDA-MB-468 and HCT-116 to BSO alone (Figure 9C,D). Guided by MTT results, cells were incubated with BSO for 24 h at concentrations 10, 20, 100, and 200 μM, and GSH activity was measured (Figure 9E,F). In conclusion, concentrations of 10 and 200 µM BSO were selected to deplete GSH in MDA-MB-468 and HCT-116, respectively, before TQ introduction. These concentrations similarly depleted GSH activity by ~88% of the values obtained from untreated controls (i.e., GSH activity is depleted to ~12% of its control values) in both MDA-MB-468 and HCT-116 without, or minimally, affecting cell viability. Then, a modified MTT assay was performed adopting BSO concentrations that depleted GSH but did not affect MDA-MB-468 or HCT-116 cell growth and viability, in which cells were pre-incubated with BSO. After 24 h, TQ was introduced, and cells were incubated for an additional 72 h. Cell growth and viability were determined by MTT assays. Concurrently, a similar experimental set for each cell line was established where cells were treated with TQ only, without BSO pre-treatment and continuous incubation. Figure 10 and Appendix A demonstrate that GSH-depleted cells exhibited greater sensitivity to TQ than non-GSH-depleted cells. TQ GI_50_ in HCT-116 decreased significantly from ~13.0 µM to ~3.5 µM in GSH-depleted cells. In MDA-MB-468, TQ GI_50_ slightly but not significantly decreased following GSH depletion from 1.0 to 0.8 µM.

### 2.9. Synthesis of Thymoquinone Analogue (TQ1) Could Bypass GSH Inactivation

GSH is known to reduce (and inactivate) TQ into glutathionyldihydro-TQ by a non-enzymatic reaction that involves one-step, two-electron transfer (Michael addition). GSH depletion could increase (HCT-116) cancer cells’ sensitivity to TQ. Hence, we sought to synthesise a TQ analogue (TQ1) that can bypass TQ inactivation by GSH. Benzylamine was used as a source of nucleophilic nitrogen that can attack carbon 3 of TQ (Figure 11). The TQ1 structure ((3-(benzylamino)-5-isopropyl-2-methylcyclohexa-2,5-diene-1,4-dione) was confirmed using 1-dimensional ^1^H NMR spectroscopy (400 MHZ, chloroform-*d*) (Appendix A). 

### 2.10. TQ1 Displayed Decreased Growth-Inhibitory Actions Compared to Thymoquinone

TQ1 antiproliferative activity was tested in both MDA-MB-468 and HCT-116 cells using the MTT assay (Figure 12). Conversely to our hypothesis, TQ1 was significantly (*p* ≤ 0.05) less potent as a growth-inhibitory agent compared to TQ in MDA-MB-468, as its GI_50_ increased to 13.5 ± 2.7 µM. However, TQ1’s GI_50_ was comparable to TQ in HCT-116 (14.4 ± 1.5 µM).

## 3. Discussion

TQ is the main biologically active constituent of *Nigella sativa*; many researchers have described TQ’s ability to inhibit the growth of cancer cells [4]. In this work, we investigated the resistance mechanisms of cancer cells to TQ. Initially, TQ’s growth-inhibitory activity was tested using MTT assays against a panel of human-derived carcinoma cells from the breast, colon, lung, and pancreas (Figure 2A, Table 1). MDA-MB-468 and T-47D were the most sensitive to TQ’s anti-proliferative actions compared to other cell lines (A549, HCT-116, HT-29, MCF-7, MIAPaCa-2). Moreover, MDA-MB-468 and T-47D were ~10-fold more sensitive than our normal cell line model (MRC-5 fibroblasts); hence, TQ exhibited selectivity. The sensitivity of MDA-MB-468 and T-47D to TQ is concordant with previous studies [5,21]. Additionally, TQ efficiently suppressed MDA-MB-468 and T-47D colony-forming ability (Figure 2B). Accordingly, individual cells were unable to resist TQ’s cytotoxic effects and either failed to survive and/or proliferate to form progeny colonies, after just 24 h exposure. Conversely, clonogenic assays (corroborating MTT data) revealed that other carcinoma cell lines—A549, HCT-116, HT-29, MCF-7, and MIAPaCa-2—demonstrated greater resistance to TQ. Thus, these cells survived short-term exposure and recovered proliferative capacity to form colonies at 1 µM TQ, but they failed to do so at 5 and 10 µM. Notably, colony formation in the more TQ-resistant cells was thwarted at lower concentrations compared to growth inhibition detected in MTT assays, indicating the cytotoxic (rather than cytostatic) potential of TQ. Nevertheless, MTT and clonogenic assays concur to indicate that MDA-MB-468 and T-47D proliferation and viability were significantly compromised by TQ at lower concentrations compared to other cell lines.

Flow-cytometric analyses allowed the detection of cell-cycle perturbations induced by TQ. Upon intercalating DNA, PI becomes potently fluorescent, allowing quantification of fluorescently stained cellular DNA content. The evolution of sub-diploid pre-G1 (<2N-DNA) events, especially in MDA-MB-468 and T-47D populations (but also HCT-116 and MCF-7), indicated apoptosis induction. As a result, the % events in other cycle phases substantially decreased, especially at 10 µM TQ for MDA-MB-468 and T-47D, and 20 µM TQ for HCT-116 and MCF-7 cells. Higher TQ concentrations were required to achieve equal perturbations in MCF-7 and HCT-116 cell cycles. Intriguingly, recovery from TQ’s actions was apparent following longer incubation (72 h), evidenced by decreased sub-diploid populations and reduced changes in G1, S, and G2/M events. The recovery of MCF-7 and HCT-116 cells could be a consequence of TQ-inactivation (or the emergence of resistance to TQ). Furthermore, annexin V-FITC/PI studies of MDA-MB-468 and HCT-116 confirmed apoptosis induction (Figure 4). MDA-MB-468 showed concentration-dependent increases in early-apoptotic populations. Membrane asymmetry of these cells is lost, exposing phosphatidylserine (PS) to the outer membrane surface (an early-apoptotic process); PS consequently binds to Annexin V-FITC. With prolonged exposure, most apoptotic events shifted into the late-apoptotic quadrant. These findings indicate that the membrane integrity of these cells was lost, exposing DNA (a late event in the apoptosis pathway) and allowing intercalation by PI. HCT-116 also displayed increased apoptotic events but to a reduced extent. Interestingly, there were more late-apoptotic than early-apoptotic cells; after 72 h, no further apoptotic events were observed. Hence, TQ initially induced transient irreparable damage earlier (within 24 h) in HCT-116 cells before its actions were neutralized, hindering apoptosis, possibly by the emergence or activation of resistance mechanisms. 

Many quinones are known to increase ROS production, an apoptosis-inducing anticancer mechanism [23]. In this study, we confirmed TQ’s ability to generate ROS (in a concentration-dependent fashion) as an early event in MDA-MB-468 and T-47D cells. However, ROS levels declined or returned to normal following 24 h treatment. ROS production could be responsible for, or contribute to, TQ’s apoptotic properties, as has been reported for many natural-product-derived anticancer agents [25]. One mechanism through which ROS could contribute to apoptosis is via causing irreparable DNA damage, leading ultimately to mitotic catastrophe. If cells bypassed apoptosis through activating cell-cycle checkpoints, they would favour asymmetric division (aneuploidy) [26]. Indeed, we observed that TQ caused early aneuploidy in MDA-MB-468 cells in a similar time frame to that of ROS generation (6 h, 12 h). Moreover, the % of aneuploid cells declined as ROS levels decreased after 24 h. Aneuploid cells are the main consequence of mitotic slippage; i.e., these cells escape cytokinesis, re-enter into another cycle, and then either pass directly into apoptosis or continue another division round. On the other hand, the emergence of aneuploid cells after mitotic slippage might render these cells more resistant to anticancer agents [27]. Resistance to oxidative stress is conferred by the enzyme NQO1, which catalyses quinones reduction into hydroquinones and the ROS-scavenger GSH. These proteins allow cancer cells to outmanoeuvre increased ROS and can ultimately reduce the potency of ROS-inducing anticancer agents. We found that NQO1 is actively expressed in cell lines with decreased sensitivity to TQ (A549, HCT-116, HT-29, MCF-7, and MIAPaCa-2, as confirmed by MTT and clonogenic assays). NQO1 can allow these cells to neutralize TQ-induced oxidative stress and survive. However, sensitive MDA-MB-468 and T-47D (which showed increased ROS with TQ) do not express NQO1. Furthermore, we saw that HCT-116 cells upregulate NQO1 expression when treated with TQ, while no NQO1 expression was seen in TQ-treated MDA-MB-468. Interestingly, NQO1 is a known stabilizer of wild-type (but not mutant) p53 in many carcinoma cells, including HCT-116 [28,29,30]; in these cells, NQO1 confers a level of protection against (resistance to) TQ. In MDA-MB-468, the absence of NQO1, consistent with other studies that failed to detect its expression in these cells [21,31], may underlie the enhanced sensitivity to TQ shown by MDA-MB-468 cells. Interestingly, and consistent with our observations, Kalo E et al. (2012) reported that carcinoma cells expressing mutant-p53 (such as MDA-MB-468) failed to induce NQO1 expression in response to oxidative stress [32]. Hence, one would expect to see increased ROS in MDA-MB-468 (due to delayed ROS neutralisation). However, the ROS-scavenger GSH might negatively impact TQ’s activity in these cells. Indeed, we found that GSH is active in both untreated MDA-MB-468 and HCT-116 cells, and its activity is not affected when cells are treated with TQ at concentrations of 1 and 10 µM, respectively (Figure 9). However, significant decreases in GSH activity were seen in MDA-MB-468 using 5 µM and HCT-116 using 20 µM TQ. It was noted that lower BSO concentrations are required to supress GSH in MDA-MB-468 compared to HCT-116 cells. In fact, mutant-p53 was shown to diminish glutathione synthesis via supressing an important GSH-ROS system’s controller gene namely SLC7A11, rendering mutant-p53 tumours susceptible to oxidative damage [33,34]. In conclusion, both NQO1 and GSH conferred protection against TQ-induced ROS in HCT-116, and possibly other less sensitive cell lines. Alternatively, MDA-MB-468 cells appear to rely mainly on GSH for protection against ROS. NQO1 and GSH (in addition to their ROS-neutralizing actions) contribute to TQ oxido-reduction cycling [20]. They can reverse the pro-oxidant action of TQ’s semiquinone radicals (produced by its one-step one-electron reduction). NQO1 and GSH can convert TQ directly into antioxidants (thymohydroquinone and glutathionyldihydro-TQ, respectively) by one-step two-electron reduction (Figure 13) [20]. Since TQ can deplete GSH and block its conversion into glutathionyldihydro-TQ, one might expect increased semiquinone production. Semiquinone production is enhanced by three enzymes: NADPH CYP reductase (E1), NADH CYP-b5 reductase (E2), and NADH ubiquinone oxidoreductase (E3). Semiquinone radicals (upon undergoing another one-step one-electron reduction) can further enhance oxidative stress by generating superoxide radicals. Moreover, semiquinone radicals (alone) have been shown recently to strongly intercalate DNA and may result in DNA strand breaks and interruption of DNA replication [35].

Therefore, in this study, we used BSO to deplete GSH. BSO inhibits γ-glutamylcysteine synthetase, the enzyme required for glutathione synthesis. We found that GSH depletion significantly (*p*-value < 0.05) enhanced TQ’s activity in HCT-116 (~4-fold), while less enhancement is seen in MDA-MB-468 (~1.3-fold) compared to näive cells. These results highlight the importance of GSH depletion as a rationale to improve TQ’s actions. Interestingly, one study showed that GSH depletion by BSO could simultaneously increase NQO1 expression in vitro [36]. This means that GSH-depleting combination therapy could synergize with TQ even in cells possessing increased NQO1 expression. Consequently, we attempted to modify the TQ structure to test the hypothesis that the Michael addition (the one-step two-electron reduction reaction between TQ and GSH) could be bypassed. According to our hypothesis, TQ reduction would shift to a one-step one-electron reaction and produce more semiquinone radicals. GSH is a strong nucleophile (Michael-donor), with the –SH group responsible for its nucleophilicity. The –SH can attack carbon-3 of TQ (Michael-acceptor), leading to the production of glutathionyldihydro-TQ (Appendix A). To prevent the strong nucleophilic attack of GSH, we used the less nucleophilic nitrogen (as a Michael-donor) of benzylamine, which can similarly attack carbon-3 of TQ. The resulting analogue was termed TQ1 (3-(benzylamino)-5-isopropyl-2-methylcyclohexa-2,5-diene-1,4-dione) (Figure 11). In disagreement with our hypothesis, TQ1’s activity was significantly (*p* ≤ 0.05) less potent against MDA-MB-468 cells compared to TQ; however, HCT-116 cells retained similar sensitivity to TQ1 as to TQ. A possible explanation could be that the enzymes E1, E2, and E3 (Figure 13) failed to convert TQ1 into semiquinone radicals and produce ROS that we suggested were responsible for TQ-induced aneuploidy and apoptosis. Such failure could be derived from the importance of TQ’s carbon-3, which seems to be essential for TQ’s conversion to a semiquinone.

In conclusion, we have demonstrated that ROS generation, aneuploidy, and cell-cycle perturbation are early events triggered by TQ that may contribute to this natural product’s anticancer activity. We confirm that NQO1 expression and GSH activity confer protection against TQ activity, but analogue TQ1, designed to thwart resistance mechanisms, failed to enhance the activity of TQ.

## 4. Materials and Methods

### 4.1. Cell Culture

All cell lines were procured from American Type Culture Collection (ATCC), grown and maintained in RPMI-1640 medium supplemented with 10% foetal bovine serum (FBS). Cells were subcultured twice a week to sustain logarithmic growth and incubated in a humidified atmosphere containing 5% CO_2_ at 37 °C. After ≤30 passages and to reduce phenotypic-genotypic drift, new lower passages were reintroduced from cell stocks that were cryopreserved at −180 °C. TQ (Sigma-Aldrich, Gillingham, UK) top stocks were kept frozen at −80 °C as 20 mM/10 µL aliquots in DMSO (Sigma-Aldrich, Gillingham, UK), which were thawed promptly before use. 

### 4.2. MTT Assay

Cell viability was determined using an MTT (3-(4,5-dimethylthiazol-2-yl)-2,5-diphenyltetrazolium bromide, Alfa-Aesar, Lancashire, UK) colourimetric assay. Briefly, cells were seeded into 96-well plates (3 × 10^3^ cells/well) and allowed to attach overnight before TQ introduction. Serial dilutions (0.02 μM–200 μM, *n* ≥ 4) were freshly prepared in a culture medium. Cell viability was measured twice (at the time of treatment introduction and after 72 h) by MTT addition. Plates were then incubated for 3 h, allowing metabolic reduction of yellow MTT into insoluble purple formazan crystals, which were solubilised in 150 μL DMSO after the supernatants’ aspiration. Data were collected at absorbance 570 nm in a plate reader (PerkinElmer Life and Analytical Sciences, Buckinghamshire, UK). TQ concentrations required to inhibit growth by 50% (GI_50_) were calculated using non-linear regression analysis. 

### 4.3. Clonogenic Assay

Clonogenic assays were used to assess single cells’ ability to survive, proliferate, and form colonies following brief exposure to TQ. Cells (400/well) were seeded into 6-well plates in a 2 mL medium. After 24 h incubation, TQ was introduced (1, 5, and 10 µM), while medium alone was added to controls. Following 24 h treatment, medium containing TQ was gently aspirated, cells were washed (2× sterile PBS, 37 °C) before fresh TQ-free medium was introduced, and plates were re-incubated for 7–9 days. The experiment was concluded when cells in control wells had formed colonies consisting of >50 cells. Wells were washed with PBS, fixed with methanol (100%; 15 min), stained with methylene blue (0.5%; 15 min), washed (dH_2_O), and air-dried, and stained colonies were then counted.

### 4.4. Cell-Cycle Analysis

The method used is adapted from that described by Nicoletti et al. (1991) [37]. Cells were seeded and allowed to attach overnight in 6-well plates (1.0 × 10^5^ and 0.5 × 10^5^ cells/well for 24 h and 72 h exposures, respectively). Following treatment, floating cells were collected, attached cells were trypsinised, and cells were pooled together in FACS tubes. Samples were washed (PBS; 4 °C) before being pelleted by centrifugation (1200 rpm; 5 min; 4 °C). Cells were re-suspended in fluorochrome solution (50 μg/mL PI, 100 µg/mL ribonuclease-A, 0.1% triton-X-100, 0.1% sodium citrate). Cell-cycle analyses were performed using an FC-500 Beckham-Coulter flow cytometer. Data were analysed using FlowJo software (version 10, OR, USA).

### 4.5. Annexin-V/PI Apoptosis Assay

Cells were seeded at densities of 0.5–0.75 × 10^5^ per well in a 6-well plate, incubated overnight, and treated the following day. After 24 h and 72 h, cells were harvested, washed (PBS; 4 °C), and transferred with medium containing detached cells into FACS tubes. After centrifugation (1200 rpm; 5 min; 4 °C) and aspiration of supernatants, cells were re-suspended in 5 μL annexin-V-FITC plus 100 μL 1× annexin-V binding buffer (BD Pharmingen). Samples were briefly vortexed and incubated (protected from light for 15 min, 25 °C). PI (10 μL) plus 400 μL annexin-V binding buffer were added into each FACS tube followed by incubation for 10 min in the dark at 4 °C. Samples were analysed using Beckman Coulter FC-500 flow cytometer and EXPO32 software, and percentages of early-apoptotic (annexin-V positive (A+), PI negative (PI−)) and late-apoptotic cells (A+, PI+) were recorded.

### 4.6. Western Blotting

Following designated treatments, cells were collected and lysed using NP-40 lysis buffer containing protease and phosphatase inhibitors. Protein concentrations were measured using the Bradford assay [38]. Samples were then loaded (50 μg protein each) and proteins were separated using sodium dodecyl sulphate polyacrylamide gel electrophoresis (SDS-PAGE). By immunoblotting, proteins were transferred onto nitrocellulose membrane, probed against primary antibody (dilution 1:1000), and thereafter against secondary antibody (dilution 1:4000). Protein bands were detected using enhanced chemiluminescence (Amersham ECL Prime (Buckinghamshire, UK) and LI-COR Biosciences Ltd. C-DiGit^®^ Blot Scanner, Cambridgeshire, UK). Densitometric analyses of western blots were performed using Image Studio. Primary antibodies were procured from CST (London, UK): NQO1 and β-Actin. A secondary anti-mouse IgG/HRP antibody was purchased from Dako (Cambridgeshire, UK).

### 4.7. Reactive Oxygen Species Assay

To measure ROS production, the ROS-Glo H_2_O_2_ (Promega, Madison, WI, USA) assay was conducted according to the manufacturer’s protocol, which measures the most stable ROS-H_2_O_2_ level. Cells were plated at a density of 5 × 10^3^/70 μL medium in white-walled opaque bottom 96-well plates and incubated overnight (37 °C, 5% CO_2_). TQ or medium (10 μL) was added to treated or control wells, respectively, for an extra 6 h and 24 h. Then, 20 μL H_2_O_2_ substrate was added to each well either simultaneously with TQ in the 6 h group or after 18 h in the 24 h group. Finally, 100 μL ROS-Glo solution was added to each well, and after 20 min, relative luminescence was measured in a plate reader (PerkinElmer Life and Analytical Sciences, Buckinghamshire, UK).

### 4.8. GSH Activity Assay

To measure GSH, the GSH-Glo Glutathione Assay (Promega) assay was conducted according to the manufacturer’s protocol. Cells were plated at a density of 5 × 10^3^/180 μL medium in white-walled opaque bottom 96-well plates and incubated overnight (37 °C, 5% CO_2_). TQ or BSO (Sigma-Aldrich, Gillingham, UK) was introduced using designated concentrations for another 24 h. The medium was aspirated, and a freshly prepared 1× GSH-Glo reagent (containing luciferin-NT, glutathione S-transferase and reaction buffer) was added. Samples were mixed briefly in a shaker and incubated (30 min; 25 °C). Reconstituted luciferin detection reagent was added for another 15 min before measuring relative luminescence.

### 4.9. Synthesis of TQ1

To a stirred solution of TQ (50 mg, 0.3 mmol) in MeOH (5 mL), benzylamine (30 µL, 30 mmol) was introduced by dropwise addition. Following 72 h, the reaction mixture was concentrated onto silica gel and purified via flash column chromatography (silica gel), eluting with petroleum ether (40–60) and 20% ethyl acetate to produce a purple solid (40 mg, 54%). TQ1 structure was confirmed by nuclear magnetic resonance ^1^H-NMR spectra.

### 4.10. Statistical Analysis

Experiments were repeated at least three times, and representative experiments are shown (unless otherwise stated). Data are presented as mean ± SD (or SEM). Statistical differences between study groups were analysed using two-way ANOVA (unless otherwise stated). Dunnett’s multiple comparisons were implicated to test significance, determined as *p*-value < 0.05.

## 5. Conclusions

We confirmed TQ’s ability to inhibit the growth and clonogenic capacity of carcinoma cells, particularly breast MDA-MB-468 (triple-negative) and T-47D (oestrogen receptor-positive) cells. More importantly, we disclose in this study the following: (i) TQ induces aneuploidy in MDA-MB-468 cells promptly with ROS production; (ii) Lysates prepared from less sensitive cell lines (A549, HCT-116, HT-29 and MIAPaCa-2) expressed NQO1. HCT-116 also exhibited evident GSH activity, and prior GSH depletion significantly enhanced TQ’s anticancer activities in these cells even in the presence of NQO1; (iii) Modifying TQ at the carbon-3 position failed to enhance anticancer activity. Indeed, TQ and TQ1 are equiactive in HCT-116 cells, while the putative step in MDA-MB-468 cells that renders them more sensitive to TQ was lost and/or inhibited following structural modification. To our knowledge, this is the first work investigating changes in both NQO1 expression and GSH activity in two carcinoma cell lines following treatment with TQ. 

## Figures and Tables

**Figure 1 molecules-26-05136-f001:**
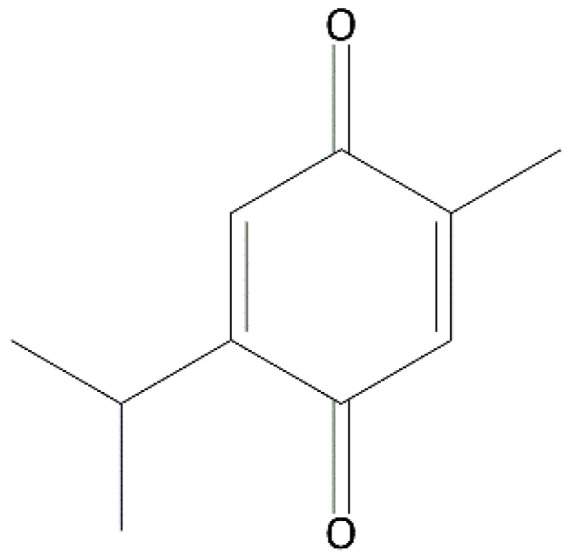
Thymoquinone Structure.

**Figure 2 molecules-26-05136-f002:**
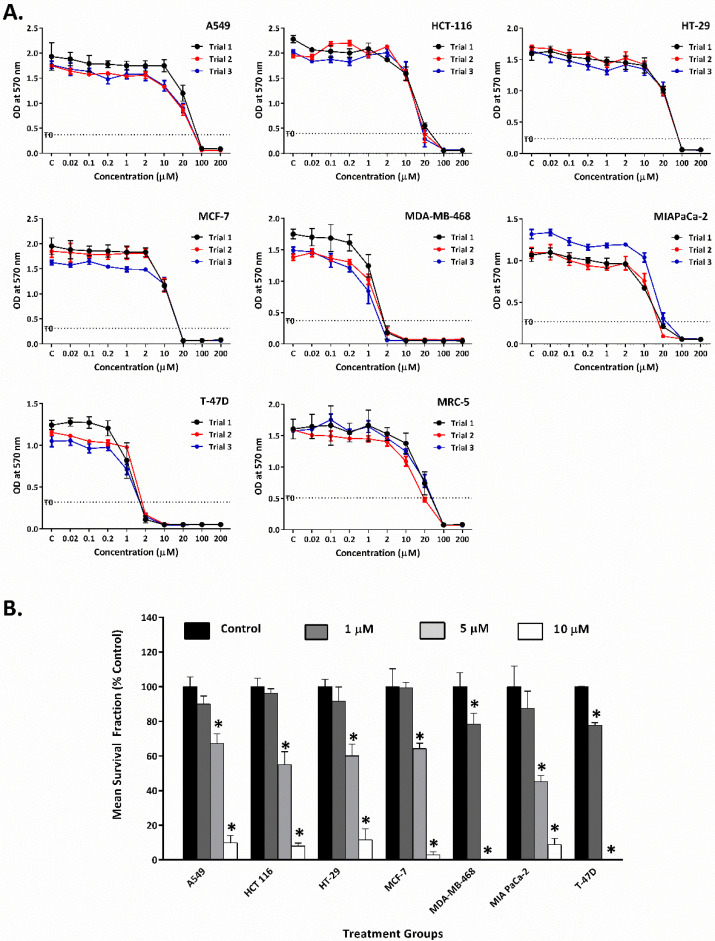
(**A**). Line graphs show TQ’s growth-inhibitory effects on A549, HCT-116, HT-29, MCF-7, MDA-MB-468, MIAPaCa-2, T-47D, and MRC-5. Each graph shows three independent MTT trials. For each graph, the T_0_ value is the average of three trials. Cells were seeded in 96-well plates (3 × 10^3^ cells/well) and treated with TQ for 72 h. No. trials ≥ 3; *n* = 4 per independent experiment. (**B**). Mean ± SD bars show TQ’s inhibition of A549, HCT 116, HT-29, MCF-7, MDA-MB-468, MIAPaCa-2, and T-47D colonies. Data presented as mean survival fraction as % of control. Asterisk indicates significant inhibition (*p* ≤ 0.05). Cells were seeded, treated with TQ (24 h), and then medium was replaced. When colonies contained ≥50 cells in control wells, colonies were fixed, stained, and counted. Plating efficiencies ranged between 20 and 35%. No. trials ≥ 3; *n* = 2 per independent trial.

**Figure 3 molecules-26-05136-f003:**
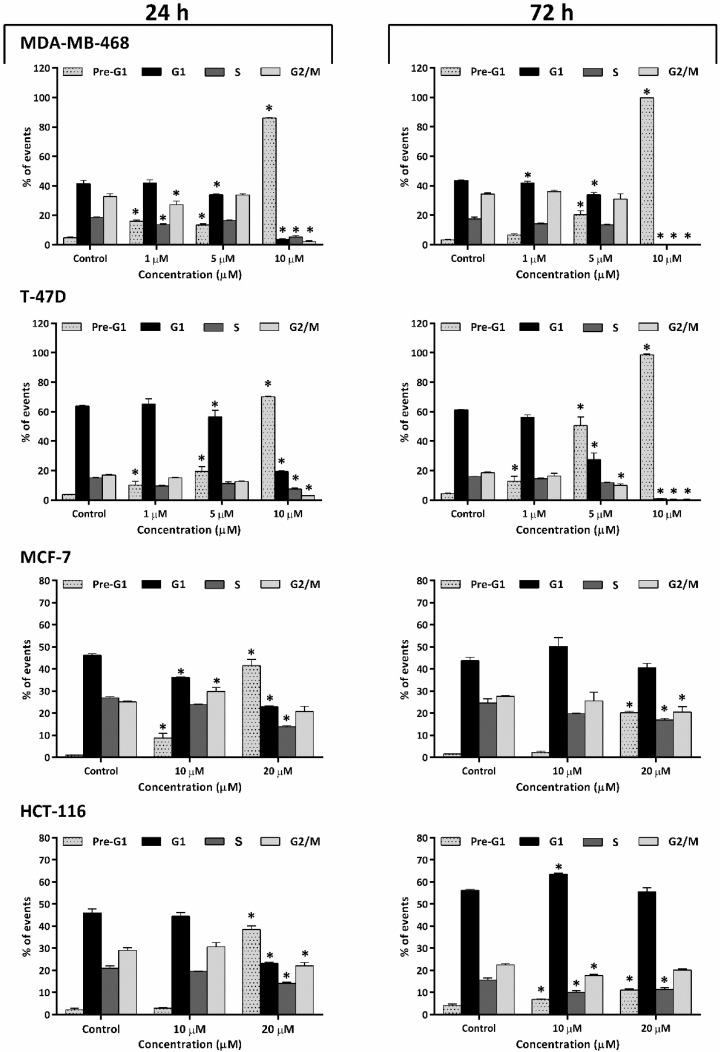
Mean ± SD bars showing % events distribution of MDA-MB-468 T-47D, MCF-7, and HCT-116 cell-cycles after 24, 72 h TQ exposures. MDA-MB-468 and T-47D treated with 1, 5, 10 μM TQ, and MCF-7 and HCT-116 treated with 10, 20 μM TQ. In former cells, TQ induced a significant concentration- and time-dependent increase in pre-G1 (<2N), with decreases in other cell-cycle phases. In the latter cells and after 24 h, TQ also induced a significant concentration-dependent increase pre-G1 (<2N), but after 72 h, fewer pre-G1 events were seen. Cells were treated, then stained with PI, and ≥20,000 events/sample were analysed. Asterisks indicate significant (*p* ≤ 0.05) changes compared to control. No. trials ≥ 3; *n* = 2 per independent experiment.

**Figure 4 molecules-26-05136-f004:**
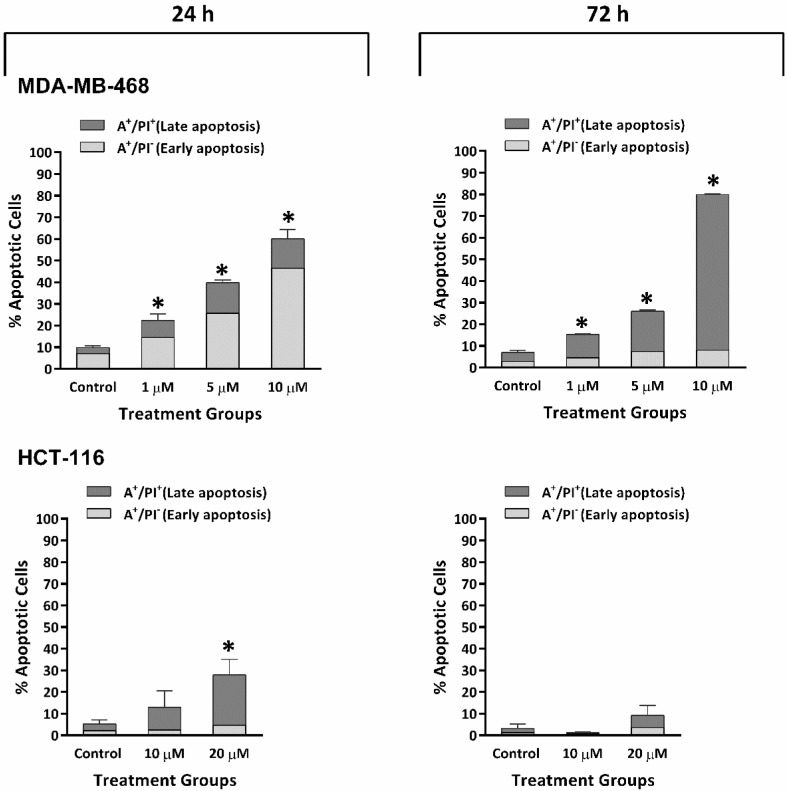
Mean ± SD bars showing annexin V/PI results of MDA-MB-468 and HCT-116 cells treated with TQ for 24, 72 h. MDA-MB-468 cells were treated with 1, 5, 10 μM TQ, and HCT-116 treated with 10, 20 μM TQ. TQ caused a significant concentration-dependent increase in apoptotic events. Samples were stained with annexin V/PI, and ≥10,000 events were detected. The percentage of apoptotic events was equal to the sum of cells undergoing early apoptosis (A+/PI−) plus late apoptosis (A+/PI+). Asterisks indicate statistically significant (*p* ≤ 0.05) increments compared to control. No. trials ≥ 3; *n* = 2 per independent experiment.

**Figure 5 molecules-26-05136-f005:**
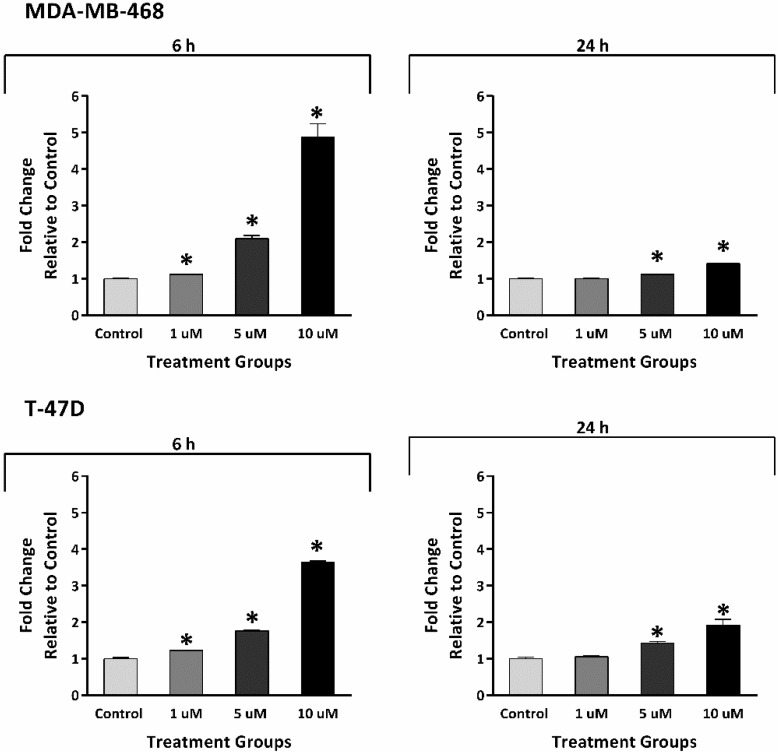
Mean ± SD bars showing a significant increase in ROS caused by TQ in MDA-MB-468 and T-47D in a concentration-dependent fashion. Higher ROS levels were seen in both cells after 6 h TQ treatments (1, 5, 10 μM) compared to the control. After 24 h, TQ (5, 10 μM) increased ROS but to reduced levels compared to 6 h treatments. No. trials ≥ 3; *n* = 2 per independent experiment. Asterisk indicates a statistically significant (*p* ≤ 0.05) fold increase relative to control.

**Figure 6 molecules-26-05136-f006:**
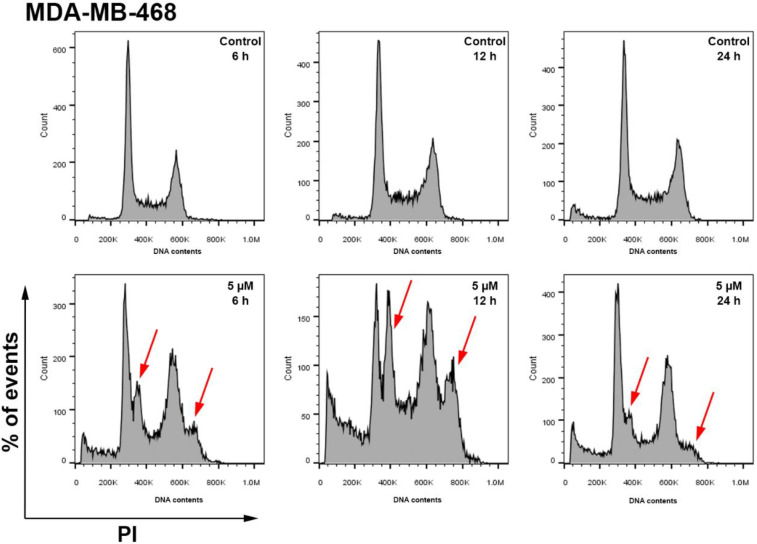
Representative histograms showing the effect of TQ (5 μM) on MDA-MB-468 cell cycle following 6, 12, 24 h exposures. TQ induced significant time-dependent increase in aneuploid cells at G1, S, and G2/M phases after 6 and 12 h with increased pre-G1. After 24 h, pre-G1 and fewer aneuploid cells were observed. Red arrows show aneuploid cells. Cells were stained with PI, and ≥20,000 events/sample were detected. No. trials ≥ 3; *n* = 2 per independent experiment.

**Figure 7 molecules-26-05136-f007:**
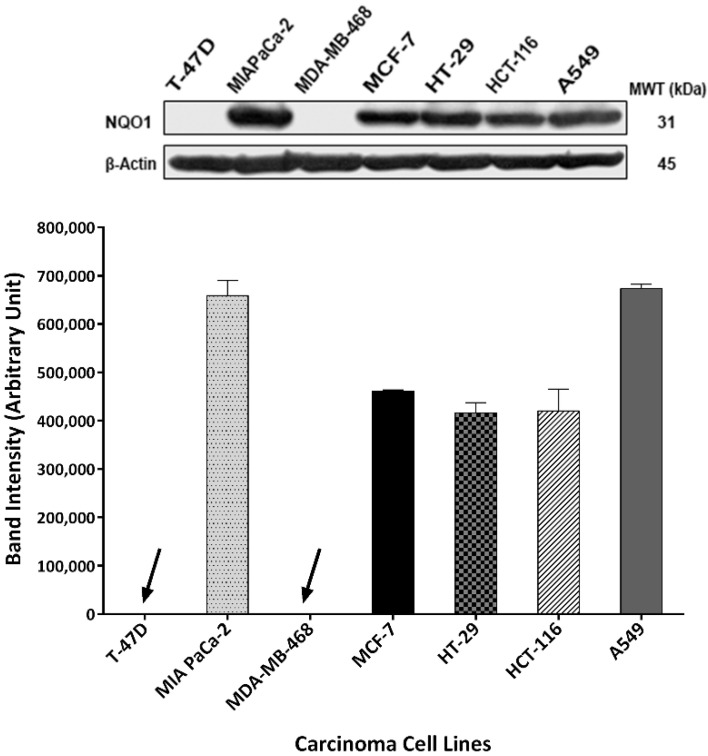
Representative western blot bands and mean ± SD bars showing NQO1 expression and band intensity in untreated A549, HCT-116, HT-29, MCF-7, MDA-MB-468, MIAPaCa-2, T-47D protein lysates. Antibodies to NQO1 and housekeeping gene β-Actin were used. NQO1 expression was seen in tested cell lines except for MDA-MB-468 and T-47D. Arrows show no detectable NQO1 expression in MDA-MB-468 and T-47D. Assay repeated three times.

**Figure 8 molecules-26-05136-f008:**
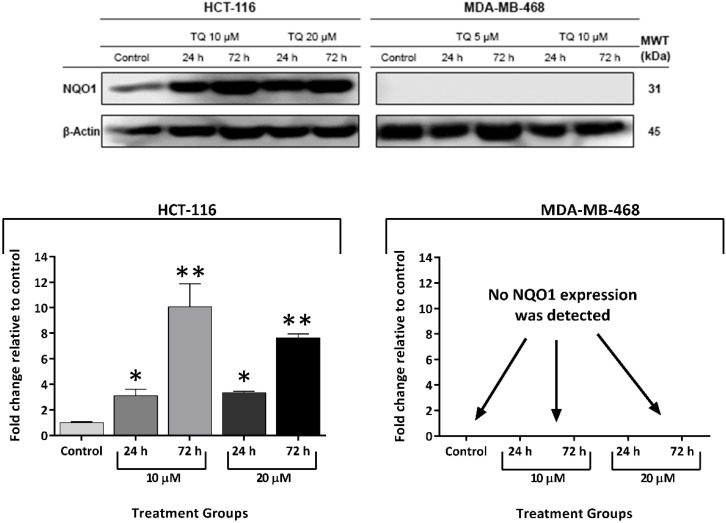
Representative western blot bands and mean ± SD bars showing NQO1 expression in HCT-116 and MDA-MB-468 protein lysates after TQ treatments (24 and 72 h) using 10, 20 μM and 5, 10 μM TQ, respectively. Antibodies to NQO1 and housekeeping gene β-Actin were used. A time-dependent increase in NQO1 was observed in HCT-116, while no NQO1 expression was seen in MDA-MB-468 lysates. Assays were repeated three times. Asterisks indicate a statistically significant (* *p* ≤ 0.05, ** *p* ≤ 0.01) change compared to the control.

**Figure 9 molecules-26-05136-f009:**
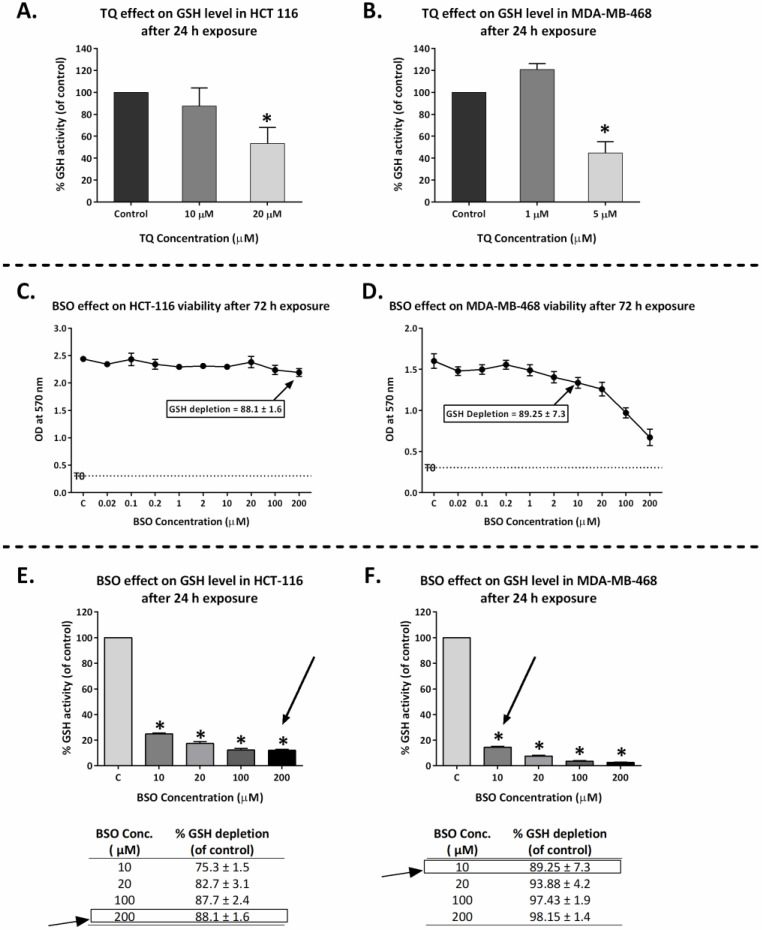
GSH depletion study in MDA-MB-468 and HCT-116 cells. (**A**,**B**): Mean ± SD bars showing GSH activity level in MDA-MB-468 and HCT-116 after 24 h TQ treatment using 1, 5 μM and 10, 20 μM, respectively. TQ significantly depleted GSH in MDA-MB-468 and HCT-116 at 5 μM and 20 μM, respectively. (**C**,**D**): Line graphs showing growth-inhibitory effects of BSO in MDA-MB-468 and HCT-116. Means ± SDs from one representative MTT trial (No. trials ≥ 3; *n* = 4 per independent experiment). Cells were seeded in 96-well plates (3 × 10^3^ cells/well) and treated with BSO for 72 h. (**E**,**F**): Mean ± SD bars shows GSH depletion by BSO in MDA-MB-468 and HCT-116 cells following 24 h exposure. Rectangles show the optimum BSO concentrations for GSH depletion with minimal effects on cell proliferation. Asterisk indicates a statistically significant (*p* ≤ 0.05).

**Figure 10 molecules-26-05136-f010:**
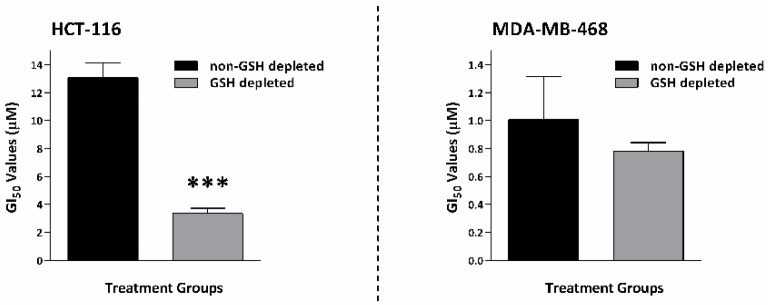
Mean ± SD bars show the effect of GSH depletion on TQ GI_50_ in MDA-MB-468 and HCT-116. GSH depletion significantly enhanced HCT-116 sensitivity to TQ. Asterisks indicate statistically significant (*p* ≤ 0.001) change compared to non-GSH depleted HCT-116. MDA-MB-468 showed slightly decreased GI_50_, which was not significant. No. trials ≥ 3; *n* = 4 per independent experiment.

**Figure 11 molecules-26-05136-f011:**
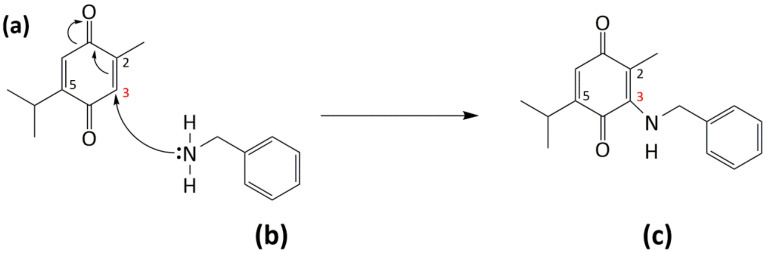
Synthesis of TQ1 (**c**) by Michael addition reaction of benzylamine (**b**) and TQ (**a**) with the less nucleophilic nitrogen attacking carbon 3 of TQ.

**Figure 12 molecules-26-05136-f012:**
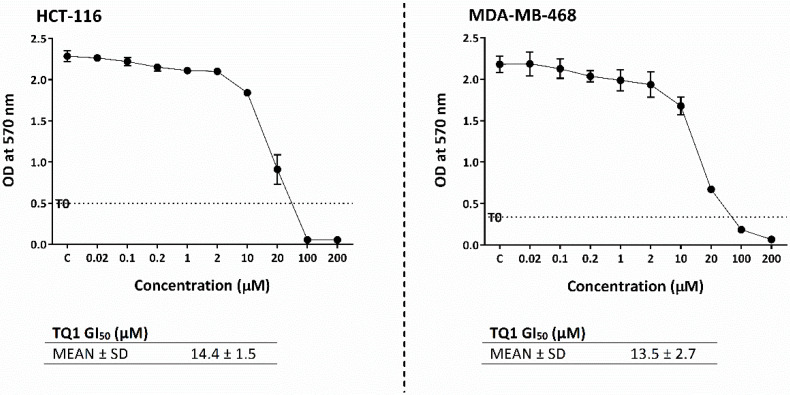
Growth-inhibitory effects of TQ1 in MDA-MB-468 and HCT-116. Each representative graph shows one independent MTT trial. Cells were seeded in 96-well plates (3 × 10^3^ cells/well) and treated with TQ1 for 72 h. No. trials ≥ 3; *n* = 4 per independent experiment.

**Figure 13 molecules-26-05136-f013:**
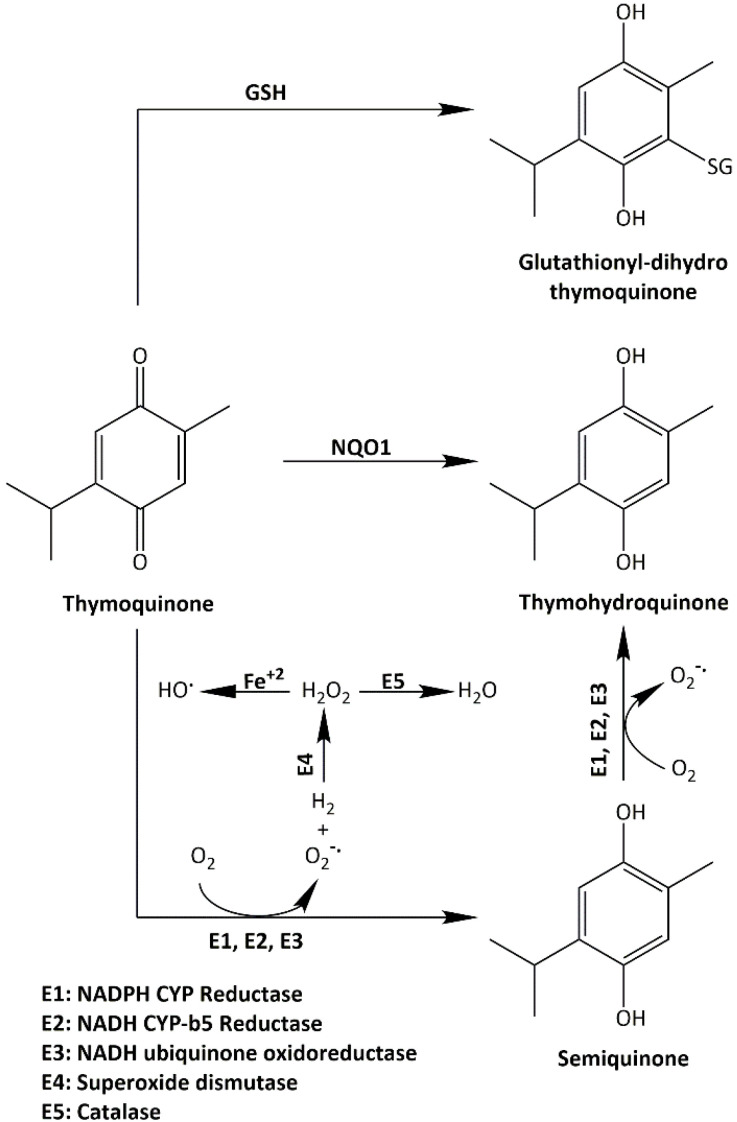
Diagram showing TQ’s oxido-reduction cycling. TQ can be converted through enzymatic reaction into thymohydroquinone either by a one-step two-electron reduction or by two-step one-electron reduction. A one-step two-electron reduction can lead to the direct formation of thymohydroquinone by NQO1. TQ may also be reduced in a non-enzymatic reaction through interaction with GSH to generate glutathionyldihydro-TQ. Alternatively, in one-electron reduction, E1, E2, and E3 catalyse TQ’s conversion into the pro-oxidant semiquinone. Thereafter, semiquinone is converted into thymohydroquinone. While thymohydroquinone and glutathionyldihydro-TQ are antioxidants, semiquinone acts as a pro-oxidant in the tumour environment. The superoxide anion produced by the oxidation of reduced TQ can be detoxified by E4 and E5. In the absence of detoxifying enzymes, which is common in numerous cancers, the increased superoxide levels can contribute to the pro-oxidant effect of TQ [20].

**Table 1 molecules-26-05136-t001:** TQ’s anti-proliferative effects on human carcinoma and fibroblast cell lines.

Cell Line	GI_50_ (µM)
Lung Carcinoma (A549)	18.8 ± 1.9
Colon Carcinoma (HCT-116)	12.7 ± 0.9
Colon Carcinoma (HT-29)	27.3 ± 3.0
Breast Carcinoma (MCF-7)	11.3 ± 1.3
Pancreatic Carcinoma (MIAPaCa-2)	12.6 ± 2.2
Breast Carcinoma (MDA-MB-468)	1.0 ± 0.2
Breast Carcinoma (T-47D)	1.5 ± 0.1
Foetal Lung Fibroblast (MRC-5)	13.4 ± 2.1

GI_50_ values were estimated from MTT assays following 72 h TQ treatment (*n* = 4 per trial) and expressed as mean ± SD from 3 independent trials.

## Data Availability

The data presented in this study are available in the article and Appendix A.

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
