# Peer review of "Concurrent Reactive Oxygen Species Generation and Aneuploidy Induction Contribute to Thymoquinone Anticancer Activity"

_molecules, 2021, doi:10.3390/molecules26175136_

Round 1

Reviewer 1 Report

Thanks for your effort. 

Author Response

Dear Reviewer,

Thank you for the opportunity to revise draft of our manuscript entitled "Concurrent reactive oxygen species-generation and aneuploidy-induction contribute to thymoquinone anticancer activity" to Molecules journal. We appreciate the time and effort that you have dedicated to providing your valuable feedback and insightful comments on our manuscript.

We are trying our efforts to improve all aspects of our manuscript. We understand your opinion regarding improving the results part. Our work adds to the current knowledge in this field – some of which will corroborate our findings, some may be in contrast to what we have found. Where we advance the field is in our design, synthesis and testing of a TQ analogue that we hypothesised would be immune to resistance mechanisms - exerted by GSH and NQO1; in the end, the hypothesis was rejected. Perhaps, there would be many to do if our hypothesis was accepted.

Thank you for your consideration.

Kind regards,

Mohammed Al-Hayali.

Reviewer 2 Report

The authors improved certain aspects of their work. However, as stated by them, their main aim was to investigate the resistance mechanisms of cancer cells to TQ. Yet, more accurately and given the results presented, they evaluated the difference in TQ effect between TQ-sensitive and TQ-resistant cancer cells. In addition, they chose MDA-MB-468 breast cancer cells as TQ-sensitive and HCT116 colon cancer cells as TQ-resistant. Why cells of different origins, and why not choosing MCF-7 as another breast cancer cell line that was TQ-resistant and NQO1 expressing? Especially when some of the TQ effects were also evaluated in the T47D breast cancer cell line (TQ-sensitive and non-expressing NQO1). According to Dai et al. (Dai, X., Cheng, H., Bai, Z., & Li, J. (2017). Breast Cancer Cell Line Classification and Its Relevance with Breast Tumor Subtyping. Journal of Cancer, 8(16), 3131–3141. https://doi.org/10.7150/jca.18457), T47D and MCF-7 are both hormone receptor-positive, HER2-negative, classified as luminal A, and evaluation of TQ effects in all breast cancer cell lines, especially the similar ones such as T47D and MCF-7, could add to the clarification of the TQ effect.

Specific comments:

  1. Why was the duration of the treatment with TQ different depending on what was investigated: cell viability (72h), colony formation (24h), cell cycle (24 and 72h), apoptosis (24 and 72 h), ROS (6 and 24h), aneuploidy (6, 12, 24h), WB-NQO1 (24 and 72h), GSH (24h), and the effect of BSO (24h) + TQ(72h)? Even the BSO treatment was different: BSO effect on viability (72h), BSO-GSH depletion (24h). Certainly, some earlier time points are set to evaluate potential preceding events such as ROS formation and aneuploidy. But, it is not clear why the effects on the viability were evaluated just after 72h treatment.

  1. The presentation of the GSH depletion in Fig. 9 is confusing. The authors measured the GSH activity and express it as a percentage of control. Yet, it is not clear what is for them % of GSH depletion (of control) because, in Fig. 9 E and F, it is presented on the graph and in the table with different values. E.g. in Fig. 9E for 200 µM BSO in the graph % of GSH depletion (of control) is about 12% while in the table % of GSH depletion (of control) is stated to be 88.1%. Therefore, the value in the table is 100% - the value from the graph. Presuming that the value on the graph is the actual % of the measured GSH activity (as later stated in the text) and since the value of the control is set at 100%, the y-axis should be renamed to GSH activity (% of control). Fig. 9 A and B should be amended accordingly.

  1. Why was not the GSH activity measured upon combined BSO+TQ treatment?

  1. In the Discussion, the authors state: “These results highlight the importance of GSH-depletion as a rationale to improve TQ`s actions. Interestingly, one study showed that GSH-depletion by BSO could simultaneously increase NQO1 expression in vitro [36]. This means that GSH-depleting combination therapy could synergize with TQ even in cells possessing increased NQO1 expression.” Since BSO can increase NQO1 expression and they showed that TQ also increases NQO1 (more after 72h) expression in HCT116 cells, and given that Sutton et al. (Sutton KM, Doucette CD, Hoskin DW. NADPH quinone oxidoreductase 1 mediates breast cancer cell resistance to thymoquinone-induced apoptosis. Biochem Biophys Res Commun. 2012 Sep 28;426(3):421-6. doi: 10.1016/j.bbrc.2012.08.111. Epub 2012 Aug 30. PMID: 22960073) have shown that NQO1 is responsible for TQ-resistance in MCF-7 cells, the authors are suggested to evaluate the expression of NQO1 after BSO+TQ treatment and viability upon 24h treatment. And as previously suggested include MCF-7 and T47D cells.

Author Response

Dear Reviewer,

Thank you for the opportunity to revise draft of our manuscript entitled "Concurrent reactive oxygen species-generation and aneuploidy-induction contribute to thymoquinone anticancer activity" to Molecules journal. We appreciate the time and effort that you have dedicated to providing your valuable feedback and insightful comments on our manuscript. Here is a point-by-point response to your comments and concerns.

Comments for reviewer 2:

  • Comment: The authors improved certain aspects of their work. However, as stated by them, their main aim was to investigate the resistance mechanisms of cancer cells to TQ. Yet, more accurately and given the results presented, they evaluated the difference in TQ effect between TQ-sensitive and TQ-resistant cancer cells. In addition, they chose MDA-MB-468 breast cancer cells as TQ-sensitive and HCT116 colon cancer cells as TQ-resistant. Why cells of different origins, and why not choosing MCF-7 as another breast cancer cell line that was TQ-resistant and NQO1 expressing? Especially when some of the TQ effects were also evaluated in the T47D breast cancer cell line (TQ-sensitive and non-expressing NQO1). According to Dai et al. (Dai, X., Cheng, H., Bai, Z., & Li, J. (2017). Breast Cancer Cell Line Classification and Its Relevance with Breast Tumor Subtyping. Journal of Cancer, 8(16), 3131–3141. https://doi.org/10.7150/jca.18457), T47D and MCF-7 are both hormone receptor-positive, HER2-negative, classified as luminal A, and evaluation of TQ effects in all breast cancer cell lines, especially the similar ones such as T47D and MCF-7, could add to the clarification of the TQ effect.

Response: True, we agree with this observation. We added the word “considered” to the abstract to further clarify what we have done. Please refer to – page number 11, Abstract, and line (12)

Regarding the cell line query – we were not investigating specifically breast cancer. The `selected` cell lines for further study represent models only. It is however a fair comment and future work could focus upon breast cancer phenotypes that display differential sensitivity / resistance to TQ. It does not however negate the research findings described in the manuscript. Moreover, we saw pre-G1 in cell-cycle analysis which is an indication for apoptosis, based on that we selected cell lines with intact apoptosis mechanism. Considering this point, MCF-7 is known to be caspase-3 deficient cells and this why its excluded.

Specific comments:

  • Comment 1: Why was the duration of the treatment with TQ different depending on what was investigated: cell viability (72h), colony formation (24h), cell cycle (24 and 72h), apoptosis (24 and 72 h), ROS (6 and 24h), aneuploidy (6, 12, 24h), WB-NQO1 (24 and 72h), GSH (24h), and the effect of BSO (24h) + TQ(72h)? Even the BSO treatment was different: BSO effect on viability (72h), BSO-GSH depletion (24h). Certainly, some earlier time points are set to evaluate potential preceding events such as ROS formation and aneuploidy. But it is not clear why the effects on the viability were evaluated just after 72h treatment.

Response: Thank you for raising this point. The two assays - clonogenic and MTT -investigate different endpoints. The clonogenic assay looks at the ability of cells to survive brief challenge and retain proliferative capacity to repopulate tumours. In this way it might be considered an in vitro mimic of chemotherapy post-surgery, or an in vitro examination of whether circulating tumour cells could survive brief exposure and produce metastatic tumours. The body will try to eliminate or metabolise/deactivate drugs and so only brief exposures of tumour cells to chemotherapy agent are anticipated. The MTT assay with continuous exposure is often used as an initial screen to determine whether a test agent has antitumour / growth inhibitory / cytotoxic potential.

Cell cycle and apoptosis assays were performed following 24 h and 72 h exposure to be consistent with MTT and clonogenic assays – to understand how the individual cells may be responding following these exposure periods.

Our interesting results (particularly cell cycle) and subsequent literature reviews led to examination of cell cycle and ROS at 6 h – in the knowledge that ROS generation is an early event and important signal transducer.

Again the 72 h is in some ways an arbitrary time/endpoint – from our initial screen – against which we wanted to compare the impact of GSH on TQ activity.

  • Comment 2: The presentation of the GSH depletion in Fig. 9 is confusing. The authors measured the GSH activity and express it as a percentage of control. Yet, it is not clear what is for them % of GSH depletion (of control) because, in Fig. 9 E and F, it is presented on the graph and in the table with different values. E.g. in Fig. 9E for 200 µM BSO in the graph % of GSH depletion (of control) is about 12% while in the table % of GSH depletion (of control) is stated to be 88.1%. Therefore, the value in the table is 100% - the value from the graph. Presuming that the value on the graph is the actual % of the measured GSH activity (as later stated in the text) and since the value of the control is set at 100%, the y-axis should be renamed to GSH activity (% of control). Fig. 9 A and B should be amended accordingly.

Response: Correct, thank you for pointing this out, we have accordingly revised Fig. 9 and the y-axis is renamed to GSH activity (% of control) for A, B, E and F figures accordingly.

  • Comment 3: Why was not the GSH activity measured upon combined BSO+TQ treatment?

Response: This is a good point. Actually, a part of our plan was to measure GSH activity upon combining BSO and TQ treatment simultaneously with measuring GSH activity after TQ1 treatment. This means we are going to use the same passages of HCT-116 and MDA-MB-468 which will give more consistent results. This step was supposed to be done after we test the activity of TQ1 to continue our work. Considering our negative results and given we need to order a new GSH activity detection kit; GSH activity measurement was omitted.

  • Comment 4: In the Discussion, the authors state: “These results highlight the importance of GSH-depletion as a rationale to improve TQ`s actions. Interestingly, one study showed that GSH-depletion by BSO could simultaneously increase NQO1 expression in vitro [36]. This means that GSH-depleting combination therapy could synergize with TQ even in cells possessing increased NQO1 expression.” Since BSO can increase NQO1 expression and they showed that TQ also increases NQO1 (more after 72h) expression in HCT116 cells, and given that Sutton et al. (Sutton KM, Doucette CD, Hoskin DW. NADPH quinone oxidoreductase 1 mediates breast cancer cell resistance to thymoquinone-induced apoptosis. Biochem Biophys Res Commun. 2012 Sep 28;426(3):421-6. doi: 10.1016/j.bbrc.2012.08.111. Epub 2012 Aug 30. PMID: 22960073) have shown that NQO1 is responsible for TQ-resistance in MCF-7 cells, the authors are suggested to evaluate the expression of NQO1 after BSO+TQ treatment and viability upon 24h treatment. And as previously suggested include MCF-7 and T47D cells.

Response: Thank you for raising this point. Our work adds to the knowledge in this field – some of which will corroborate our findings, some may be in contrast to what we have found. Both NQO1 and GSH have roles to play in the response of cells to TQ, as is reported in the literature and as we have also concluded. What we want to emphasize here is that GSH depletion alone can be efficient enough to improve TQ’s action in the presence of NQO1 (either basal or upregulated level) as in HCT-116 and in its absence as in MDA-MB-468. However, we did not check NQO1 expression by western blotting - at this stage - because we already confirmed that MDA-MB-468 cells do not express NQO1 which is consistent with other studies. Added to that, HCT-116 responded similarly to TQ1 and TQ and MDA-MB-468 become more resistant, so we could not go further in this work. Where we advance the field is in our design, synthesis and testing of a TQ analogue that we hypothesised would be immune to these resistance mechanisms; in the end, the hypothesis was rejected. Perhaps, there would be many to do if our hypothesis was accepted.

Thank you for your consideration.

Kind regards,

Mohammed Al-Hayali.

This manuscript is a resubmission of an earlier submission. The following is a list of the peer review reports and author responses from that submission.

Round 1

Reviewer 1 Report

The manuscript entitled "Thymoquinone concurrently induces aneuploidy and ROS generation; GSH-depletion greatly enhances its anticancer activity" by Mohammed Al-Hayali et al.

Major comments

  • As the authors mentioned many studies already performed on TQ and confirmed its action, therefore, there is lack of novelty in the current work.
  • In the abstract, the authors promised to investigate the resistance mechanisms to TQ, do you mean the drug resistance after prolonged treatment of TQ? But I didn't see any data correlating to resistance to TQ, may be the word "resistance" is not appropriate, which should be just drug sensitivity to TQ.
  • The cytotoxicity of TQ to normal cells should be studied.
  • The authors claimed that TQ induces aneuploidy in cancer cells, given that aneuploidy is one of the characteristics of cancer cells, then are the authors actually saying the cancer promoting effect of TQ?
  • At the end, many of the results are just observations and/or end results, and did not give sufficient underling mechanism to explain TQ's action to cause aneuploidy. Also, the way of the presentation is very confusing and not focused.
  • It is unclear on what basis to design the TQ analogue. As a matter of fact, the way of the design is not comprehensive at all and of course it is very likely to lose the drug potency.

Other comments

  • Typos and unfriendly mode of English can be found, which must be addressed.
  • Many of ther references in the article text displayed "Error!", which cannot tell people which references are they referring to.
  • The structure of TQ should be shown in the beginning section.

Reviewer 2 Report

In this manuscript, Mohammed Z.K. Al-Hayali et al showed that Thytmoquinone (TQ) induces aneuploidy and ROS generation. Overall, the authors showed that resistance mechanism of carcinoma cells to TQ. The manuscript could be further strengthened with a revision denoted below. 

  1. There are many places that incorrectly or inaccurately write down the manuscript such as page 2, line 63 (Reference source not found) etc.
  2. Please add more references in introduction part. Also, please explain more about Thymoquinone, Cancer and ROS.
  3. Many of papers already published about Thymoquinone in cancer cells. Please describe the difference between this paper and the existing one.
  4. Please explain why 24 hour data works better than 72 hour data in Figure 3.
  5. Please recheck the format of Molecules. Many places are not correct. For example, no acknowledgement.

Reviewer 3 Report

The authors wrote a very interesting and well-organized paper in which they explored the effect of Thymoquinone (TQ) on the growth of several carcinoma cell lines. A more detailed analysis of the TQ effect and the possible mechanism involved was investigated in highly sensitive MDA-MB-468 and less sensitive HCT-116 cell lines. Before publication, some of the issues should be cleared.

  1. Lines 68-69 The authors state: “DMSO vehicle had no effect on cancer cell growth.” What is the remaining percentage of DMSO? No effect of DMSO should be shown at least in the supplementary.
  2. In Figure 1A the authors showed representative graphs of one independent MTT trial. Why not showing all the data combined since they repeated the experiment three times?
  3. Lines 215-217- The authors state: “TQ (5 μM) significantly depleted GSH activity in MDA-MB-468 cells to ~45% of control values. TQ (20 μM) decreased GSH activity in HCT-116 to ~75% of untreated control. “ In Figure 8A they showed % Depletion. For MDA-MB-468 cells depleted GSH activity (TQ (5 μM)) is ~45% (~50% seems more precise) of control values but HCT-116 cells (TQ (20 μM)) is not ~75% of untreated control according to presented graph but ~40%.
  4. Again, the values presented in Figure 8E and 8F do not correspond with the text: “In conclusion, concentrations of 10 and 200 μM BSO were selected to deplete GSH in MDA-MB-468 and HCT-116 respectively before TQ introduction. These concentrations similarly depleted GSH to ~88% of untreated control in both MDA-MB-468 and HCT-116…”
  5. In the Discussion the authors state: “We found that GSH-depletion significantly enhanced TQ`s activity in HCT-116 (~4-fold) and MDA-MB-468 (~1.3-fold) cells compared to näive cells.” However, GSH-depletion does not significantly enhance TQ`s activity in MDA-MB-468 cells.
  6. The authors should be more clear concluding e.g. (i) TQ selectively inhibits carcinoma cell growth and their clonogenic capacity.
  7. In MM, Western Blotting should contain more information: what was used as lysis buffer?; how much protein was loaded?; what were the antibody dilutions?; what chemiluminescence reagent was used and how was blots visualized?
  8. Lines 63, 70, 96, 110, 125, 134, 150, 167, 184, 188, 190, 215, 238, 240, 250, 269, 277, 290, 296, 359, 368, 394, 398 - Error! Reference source not found. Hence, reference list probably is not as it should be.
  9. Some typos are present in the text.